# Language Modeling on Location-Based Social Networks

Juglar Diaz *, Felipe Bravo-Marquez 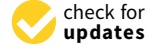 and Barbara Poblete

Department of Computer Science, University of Chile & IMFD, Santiago 8370456, Chile;
fbravo@dcc.uchile.cl (F.B.-M.); bpoblete@dcc.uchile.cl (B.P.)
* Correspondence: judiaz@dcc.uchile.cl

**Abstract:** The popularity of mobile devices with GPS capabilities, along with the worldwide adoption of social media, have created a rich source of text data combined with spatio-temporal information. Text data collected from location-based social networks can be used to gain space–time insights into human behavior and provide a view of time and space from the social media lens. From a data modeling perspective, text, time, and space have different scales and representation approaches; hence, it is not trivial to jointly represent them in a unified model. Existing approaches do not capture the sequential structure present in texts or the patterns that drive how text is generated considering the spatio-temporal context at different levels of granularity. In this work, we present a neural language model architecture that allows us to represent time and space as context for text generation at different granularities. We define the task of modeling text, timestamps, and geo-coordinates as a spatio-temporal conditioned language model task. This task definition allows us to employ the same evaluation methodology used in language modeling, which is a traditional natural language processing task that considers the sequential structure of texts. We conduct experiments over two datasets collected from location-based social networks, Twitter and Foursquare. Our experimental results show that each dataset has particular patterns for language generation under spatio-temporal conditions at different granularities. In addition, we present qualitative analyses to show how the proposed model can be used to characterize urban places.

**Keywords:** spatio-temporal text data; location-based social networks; language models

## 1. Introduction

Social networks play a crucial role nowadays in modern societies. From interests and reviews to preferences and political opinions, it is imprinted in our everyday life. Social networks such as Instagram, Facebook, Twitter, and Foursquare allow users to share text data with spatio-temporal information (a timestamp and geo-coordinates). We refer to these social networks as location-based social networks (LBSN). Text data generated on location-based social networks is a set of records representing "where", "when" and "what", in which the "where" means a location's latitude–longitude geo-coordinates, the "when" is a timestamp, and the "what" is the textual content.

Understanding patterns of spatio-temporal textual data generated on LBSN can help us understand human mobility patterns [1,2] or when and where popular social activities take place [3–5] in urban environments. In addition, spatio-temporal textual data from LBSN has been successfully used to detect real-world events such as earthquakes [6,7] or to predict events such as civil unrest [8]. A better understanding of this type of data could be beneficial in a wide range of scenarios. For instance, the STAPLES Center is a multipurpose arena in Los Angeles, California that holds different humans activities such as sporting events and concerts. Using "STAPLES Center" to annotate this location could fail to reveal the complete purpose of the place; while using data from a LBSN could discover spatio-temporal nuances of the human activities that take place on points of interest such as this.

One challenge related to modeling this kind of data is its multi-modality. Timestamps, geo-coordinates, and textual data exhibit different magnitudes and representations schemes, which makes it difficult to combine them effectively. Timestamps and geo-coordinates are continuous variables while the text is a sequence of discrete items and is usually represented using vector spaces.

An additional challenge is associated with the individual representation of each type of variable. Previous approaches (see Section 2) for modeling how text is generated in a spatio-temporal context use a single granularity representation for time or space: either using hand-crafted discretizations, automatic models such as clustering algorithms, or probabilistic models. Spatio-temporal patterns for text data generation should capture patterns at different granularities such as hours, weeks, months, and years, for time or blocks, neighborhoods and cities, for space. When considering the textual data, previous works have modeled the text following a bag-of-words approach (see Section 2), ignoring the sequential structure of texts.

The research question that guides this work is whether modeling time and space at different granularities along with the sequential structure of texts can improve the modeling of spatio-temporal conditioned text data. The main contributions of our current work are as follows:

1.  Propose a spatio-temporal conditioned neural language model architecture that represents time and space at different granularities and captures the sequential structure of texts. By modeling time and space at different granularities, the proposed architecture is adaptable to the specific characteristics of each data source. This has proven to be paramount according to our experiments over two LBSN datasets.
2.  Perform a qualitative analysis where we show visualizations that can help to gain insights into the patterns that guide language generation under spatio-temporal conditions. By modeling time and space at different granularities, we can analyze how each granularity level weighs in the representation model. For this analysis, we conducted experiments with a Transformer-based neural network. Attention-based neural networks such as the Transformer architecture have the benefit of providing insights into the importance of components of the spatio-temporal context by visualizing the attention weights.

*Roadmap*

This document is organized as follows. In Section 2, we provide a background of the literature relevant to this work. In the first part of the section, we describe applications that leverage spatio-temporal textual data from LBSN; after that, we delve into models that jointly represent the three variables and highlight existing drawbacks in previous approaches that need to be addressed. In Section 3, first, we provide a background on language modeling before presenting our problem formulation as a spatio-temporal conditioned language modeling task. We provide a background of neural networks for language modeling and finally describe the proposed neural language model architecture. In Section 4, we describe our experimental framework. We present the LBSN datasets used in our experiments, and we describe the evaluation metric and the experiments that we conducted to understand time and space modeling at different granularities. Finally, in Section 5, we discuss our conclusions.

## 2. Related Work

In this section, we provide an overview of the work in the literature related to this research. First, we describe the principal applications of spatio-temporal text data generated on LBSN. Later, we delve into the models for spatio-temporal text data closest to our work derived from these applications mentioned before. These works study how text is generated in a spatio-temporal context, and we focus on how they model time and space as a context for language generation.

### 2.1. Applications for Spatio-Temporal Text Data

As stated in previous sections, there are many sources of text data with spatio-temporal dimensions. Nevertheless, most of the works in the literature focus on the LBSN domain. It is the most abundant data source and easiest to acquire using APIs. The main applications that we identify in the literature are activity modeling, mobility modeling, event detection, and event forecasting. Next, we describe these applications.

#### 2.1.1. Activity Modeling

Activity modeling studies human activities in urban environments using spatio-temporal text data related to human activities. As people share information about activities they do in the everyday life, spatio-temporal text data from LBSN provides useful information about spatial and temporal patterns of human activities. Unlike static analysis of spatial data, spatio-temporal text data can discover the purpose of a visit to a point of interest that hosts multiple kinds of events. For instance, the STAPLES Center, a multi-purpose arena in Los Angeles, California holds sporting events as basketball matches but also can hold others, such as concerts. People may visit the STAPLES Center for different purposes. Using "STAPLES Center" to annotate a location record could fail to reveal the complete purpose of the location.

Works in activity modeling focus on place labeling and models that jointly represent text, time, and space. Both approaches characterize urban areas using data collected from LBSN. Given a set $R = \{r_1, \ldots, r_m\}$ of spatio-temporal text data records, place labeling finds labels that best describe PoIs, either static [9] or at different time periods [3]. Works that jointly represent text, time, and space for activity modeling allow combining the three data types in a unique representation scheme [4,10].

#### 2.1.2. Mobility Modeling

Mobility modeling using spatio-temporal text data allows us not only to know the geometric aspects of mobility human data but also the semantics: i.e., going from point $A$ at time $t_0$ to point $B$ at time $t_1$ is not as informative as going from "home" at time $t_0$ to "work" at time $t_1$ or from "work" at time $t_2$ to a "restaurant" at time $t_3$. Studying human mobility patterns have applications such as place prediction/recommendation [2,11] for individual users and trajectory pattern mining for mobility understanding in urban areas [1,12]. This information can lead to grasping the reasons that motivate people's mobility behaviors, understanding the nuances of mobility problems in urban environments and then taking effective actions to solve them.

#### 2.1.3. Event Detection

Event detection methods applied on streaming of spatio-temporal text data from LBSN allows us to detect, in real-time, geo-localized events from first-hand reporters. As defined by Allan et al. [13], an event is something that happens at a specific time and place and impacts people's lives, e.g., protests, disasters, sporting games, concerts. Some types of events that are reflected in LBSN and can be detected are earthquakes [6,7,14] or traffic congestion [15,16].

#### 2.1.4. Event Forecasting

Event forecasting methods, unlike event detection, which typically discovers events when they are occurring, predict the incidence of events in the future. The common approach is to use data from LBSN in conjunction with external sources to build prediction models. For some events such as criminal incidents [17–19] or civil unrests [8,19], predicting the exact location with as much time in advance is paramount. A common approach is to define features as indicators and train prediction models for spatial regions [17]. For civil unrest, the prediction is usually at the city level or smaller administrative regions, while for crimes and traffic events, the prediction is at a finer grain level such as neighborhoods or

blocks. The temporal variable is used to identify the changing patterns that indicate the occurrence of an event in the future.

### 2.2. Models for Spatio-Temporal Text Data

Analyzing the former applications, activity modeling can be considered the primary task. It allows answering $\langle what \rangle$ happens, $\langle when \rangle$ it happens, and $\langle where \rangle$ it happens and can be considered the basic task. For example spatial and temporal activity patterns can be used to define transition points in trajectories for mobility models; spatial and temporal activity patterns are used as features for event forecasting models, and unusual localized bursty activity is used to detect events. Next, we focus on specialized models for activity modeling. First, we describe models that detect geographical topics. Then, we describe multi-modal embedding methods for spatio-temporal text data.

#### 2.2.1. Spatio-Temporal Topic Modeling

Spatio-temporal topic modeling discovers topics related to geographical areas [20–26]. Mei et al. [20] proposed a generalization of the Probabilistic Latent Semantic Indexing [27] model, in which topics can be generated by *text* or by the combination of *timestamp* and *location*. Eisenstein et al. [21] proposed a cascading topic modeling. Words are generated by a multinomial distribution that is the mean of a latent topic model and a region topic model. Regions are latent variables that also generate coordinates. Topics are generated by a Dirichlet distribution. Regions are generated by a multinomial distribution, and coordinates are generated by a bivariate Gaussian distribution. Each region has a multinomial distribution over topics, and each topic has a multinomial distribution over keywords. Wang et al. [22] proposed LATM [22], which is an extension of Latent Dirichlet Allocation (LDA) [28] that is capable of learning the relationships between locations and words. In the model, each word has an associated location. For generating words, the model produces the word and also the location, in both cases with a multinomial distribution depending on a topic that is generated by a Dirichlet distribution. Additionally, Sizov [23] developed a model similar to the work of Wang et al. [22]. Rather than using a multinomial distribution to generate locations, they replace it with two Gaussian distributions that generate latitudes and longitudes. Yin et al. [4] studied a generative model where there are latent regions that are geographically distributed by a Gaussian. Hong et al. [24] use a base language model, a region-dependent language model, and a topic language model. Geo-coordinates are discretized into regions using clustering algorithms. Regions are generated by a multinomial distribution depending on the user and a global region distribution. Geo-coordinates are generated by the regions using multivariate Gaussian distributions. Words are generated by topics depending on the global topic distribution, the user, and the region. Ahmed et al. [25] developed a hierarchical topic model that models both document and region-specific topic distributions and additionally models regional variations of topics. Relations between the Gaussian distributed geographical regions are modeled by assuming a strict hierarchical relation between regions that is learned during inference. Finally, Kling et al. [26] proposed MGTM [26], a model based on multi-Dirichlet processes. The authors used a three-level hierarchical Dirichlet process with a Fischer distribution for detecting geographical clusters, a Dirichlet-multinomial document-topic distribution, and a Dirichlet-multinomial topic-word distribution.

#### 2.2.2. Embedding Methods

Embedding methods are distributed learned representations for discrete variables. Learned embedded representations are very popular in natural language processing [29,30] and graph node representation [31]. For spatio-temporal textual data, embedded-representations learn a joint representation for the elements of the tuple $\langle time, location, text \rangle$.

Zhang et al. [10] proposed CrossMap [10]. In CrossMap, the first step is to discretize timestamps and coordinates using Kernel Density Estimation techniques. After that, CrossMap uses two different strategies to learn the embedded representations: Recon

and Graph. In Recon, the problem is modeled as a relation reconstruction task between the elements of the tuple $\langle time, location, text \rangle$, while in Graph, the goal is to learn representations such that the structure of a graph built from the tuples $\langle time, location, text \rangle$ is preserved. In [5], CrossMap is extended to learn the embedded representation in a stream. The authors propose two strategies based on life-decay learning and constrained learning to find the representations from the streaming data. Unlike CrossMap, timestamps and geo-coordinates are discretized into hand-crafted spatial windows and temporal cells instead of Kernel Density Estimation-based clustering. Zhang et al. [10,32] proposed another extension to CrossMap, in this case, to learn representations from multiple sources. The main dataset is the set of tuples $\langle time, location, text \rangle$. Each dataset defines a graph, and the representations are learned to preserve the graph structure. Nodes representing the same entity are shared between the main graph and secondary graphs. During training, the learning process alternates between learning the embeddings for the main graph and the embeddings for the secondary datasets.

### 2.2.3. Analysis of Models That Leverage Spatio-Temporal Text Data

In Table 1, we present a summary of the works discussed in this section. Existing approaches are based on topic modeling or embedding methods. Works following the topic modeling approach are based on topic models such as Probabilistic Latent Semantic Analysis [33] or Latent Dirichlet Allocation [28] and extend the models by assigning distributions over locations to topics or by introducing latent geographical regions. Both topic models and embedding methods assume a bag-of-words approach for text modeling, which ignores the sequential structure of texts. When considering time and space modeling, each work models timestamps and geo-coordinates at a single level of granularity using hand-crafted spatial cells and temporal windows or clustering algorithms. Only Ahmed et al. [25] models hierarchy, but only for space; to the best of our knowledge, there are no studies of how representing time and space at different levels of granularity impact the modeling of text generation under spatio-temporal conditions. In addition, no work models the sequential structure of texts.

An additional problem about modeling spatio-temporal text data, which is important to mention, is the evaluation framework. Building a reference dataset in this field is complex. First, there is a temporal variable involved: this means that data should be collected for a long time. Second, data are related to a specific region: this means that using models in a new region would require collecting data from that region. We can observe (see column Dataset in Table 1) that there is no consensus about what dataset to use as a standard to establish fair evaluations between different approaches. For this reason, we decided not to amplify this issue by using a new dataset, and we develop our experiments using the most recent datasets (see Section 4.1) reported in [5,10,32].

In addition, each work models time and space with different techniques such as clustering, probabilistic models, or hand-crafted discretizations and uses different evaluation metrics suited to their proposed model. For example, works whose outcomes are classification models are evaluated using classification metrics such as Accuracy, works that produce Probability Distributions are evaluated using Perplexity, and works that propose ranking models are evaluated using Mean Reciprocal Rank. As in this work, we propose a spatio-temporal conditioned neural language model, we use as an evaluation metric Perplexity, which is a traditional language modeling evaluation metric. Using Perplexity over the generated text, because we only look at the text, allows us to disentangle the evaluation metric from how time and space are modeled.

Overall, we can conclude that existing approaches ignore two dimensions of the problem:

1. The sequential structure of language.
2. A unified model for representing time and space that leverage time and space at different granularities as context for language generation.

**Table 1.** Spatio-temporal text data modeling.

| Work | Time Representation | Space Representation | Text Representation | Integration | Dataset | Evaluation Metric |
|------|---------------------|---------------------|---------------------|-------------|---------|-------------------|
| [20] | Days in a week | City | Multinomial | Topic modeling | Blogs (2006) | - |
| [21] | - | User aggregation + Gaussian | Multinomial | Topic modeling | Twitter (2010) | Accuracy and Mean Distance |
| [23] | - | Two Gaussian | Multinomial | Topic modeling | Flickr (2010) | Accuracy |
| [22] | - | Multinomial | Multinomial | Topic modeling | News (-) | Perplexity |
| [24] | - | Clustering + Gaussian | Multinomial | Topic modeling | Twitter (2011) | Mean Distance |
| [25] | - | Hierarchical Gaussian | Multinomial | Topic modeling | Twitter (2011) | Accuracy and Mean Distance |
| [26] | - | Fisher distribution | Multinomial | Multi-Dirichlet process | Flickr (2010) | Perplexity |
| [10] | Clustering over seconds in a day | Clustering | Embedding | Multimodal embedding | Twitter (2014) Foursquare (2014) | Mean Reciprocal Rank |
| [5] | Hours in a day | Equal-sized grids | Embedding | Online multimodal embedding | Twitter (2014) Foursquare (2014) | Mean Reciprocal Rank |
| [32] | Hours in a day | Equal-sized grids | Embedding | Cross-modal embedding | Twitter (2014) Foursquare (2014) | Mean Reciprocal Rank |

## 3. Proposed Solution

In this section, we describe our proposed solution. First, we show the problem formulation, which is framed as a language modeling task. After that, we describe the proposed model for which we previously briefly overview state-of-the-art neural language model architectures. Finally, we show the discretizations of timestamps and geo-coordinates as well as the parameters selection.

### 3.1. Language Modeling

Language modeling is defined as the task of assigning a probability to a sequence of words $w$: $p(w) = p(w_0, w_1 \ldots w_{j-1}, w_j)$. State-of-the-art models for language modeling are based on neural networks. Typically, neural network language models are constructed and trained as discriminative predictive models that learn to predict a probability distribution $p(w_j/w_0, w_1 \ldots w_{j-1})$ for a given word conditioned on the previous words in the sequence. These models are trained on a given corpus of documents. The probability of a sequence of words $p(w_0 \ldots w_{j-1}, w_j)$ can be estimated with: $\prod_{i=1}^{i=j} p(w_i/w_0, w_1 \ldots w_{i-1})$.

Conditioned language modeling is defined as the task of assigning a probability to a sequence of words given a context $c$: $p(w/c) = p((w_0, w_1 \ldots w_{j-1}, w_j)/c)$. Then, the probability of each word in the sequence is computed as $p(w_j/c, w_0, w_1 \ldots w_{j-1})$. Conditioned language models have applications in multiple natural language processing tasks: for example, machine translation (generating text in target language conditioned on text in a source language), description of an image conditioned on the image, a summary conditioned on a text, an answer conditioned on a question and a document, etc. In our case, the context will be a tuple of timestamp and coordinates.

### 3.2. Problem Formulation

Given a collection of records that provide textual descriptions of a geographical area at different moments in time, our goal is to create a model capable of representing this multi-modal data. Following the traditional language modeling task formulation, we require the resulting model to assign a probability to a text given the timestamp and geo-coordinates associated with that text.

More formally, let be $H = \{r_1, \ldots, r_n\}$ a set of spatio-temporal annotated text records (e.g., a tweet). Each $r_i$ is a tuple $\langle t_i, l_i, e_i \rangle$, where $t_i$ is the timestamp associated with $r_i$, $l_i$ is a two-dimensional vector representing the location corresponding to $r_i$, and $e_i$ denotes the text in $r_i$. Given that $e_i$ is a sequence of words $w_0 \ldots w_n$, assigning a probability to $w_0 \ldots w_n$ given $\langle t_i, l_i \rangle$ can be written as $p((w_0, w_1 \ldots, w_n)/\langle t_i, l_i \rangle)$, which is an instance of the conditioned language modeling task presented in Section 3.1.

### 3.3. Neural Networks for Language Modeling

Since we propose a neural network architecture to model text generation under spatio-temporal conditions, we consider it is important to provide a background of the state-of-the-art neural network architectures for language modeling. We describe the two neural network architectures that have shown state-of-the-art results across many natural language processing tasks [34]: recurrent neural networks (RNN) and Transformer-based self-attention models.

Recurrent neural networks [35] are a family of neural network architectures that capture temporal dynamic behavior. RNN have been successfully applied to natural language processing problems such as speech recognition [36] and machine translation [37–39], among others. In the case of spatio-temporal data, they have been mostly used for mobility modeling [40–43]. In the basic architecture for an RNN, there is a vector $h$ that represents the sequence. At each timestep $t$, the model takes as input $h_{t-1}$ and the $t$-*th* element of the sequence $x_t$; then, it computes $h_t$. For language modeling, at each time step $t$, $h_t$ is used as input to a feed-forward network that predicts the next token $x_{t+1}$. The most popular architectures of RNN are the Long-Short Term Memory (LSTM) [44] and the Gated Recurrent Unit (GRU) [45]. Both variants introduce mechanisms that control the information flow between the hidden states representing the sequence.

Self-attention architectures have revolutionized the natural language processing (NLP) field with several works that followed this approach. The Transformer [46] was initially proposed for a language translation task. Later, pre-trained language models [47–49], following the self-attention model proposed by the Transformer, have improved the state-of-the-art for many NLP tasks. This approach uses positional encoding to leverage word positions and several layers of multi-head self-attention. The self-attention architecture removes the recurrent component of RNNs that limits parallelization. This allows faster training with superior quality when compared to previous models based on recurrent neural networks.

### 3.4. Model Description

Our proposed architecture consists of an end-to-end neural network for encoding spatial and temporal contexts and decoding/generating text. Our design is targeted to model the spatio-temporal context at different granularities and to make the decoding/generating component agnostic to how the encoding of the spatial and temporal contexts are instantiated.

Figure 1 shows the model's architecture. In order to feed our model with spatio-temporal textual data, some pre-processing steps are required. First, text is tokenized, timestamps are discretized into temporal windows, and geo-coordinates are discretized into spatial cells (Equation (1)). After that, discretized timestamps and discretized geo-coordinates are passed through embedding layers (Equation (2)). The embedding layer projects words, temporal windows and spatial cells into a dense representation. Each item is embedded using a look-up table, and there is a look-up table for each type of item:

temporal windows, spatial cells and words. Each item is associated with an integer that is used as an index in the corresponding look-up table.

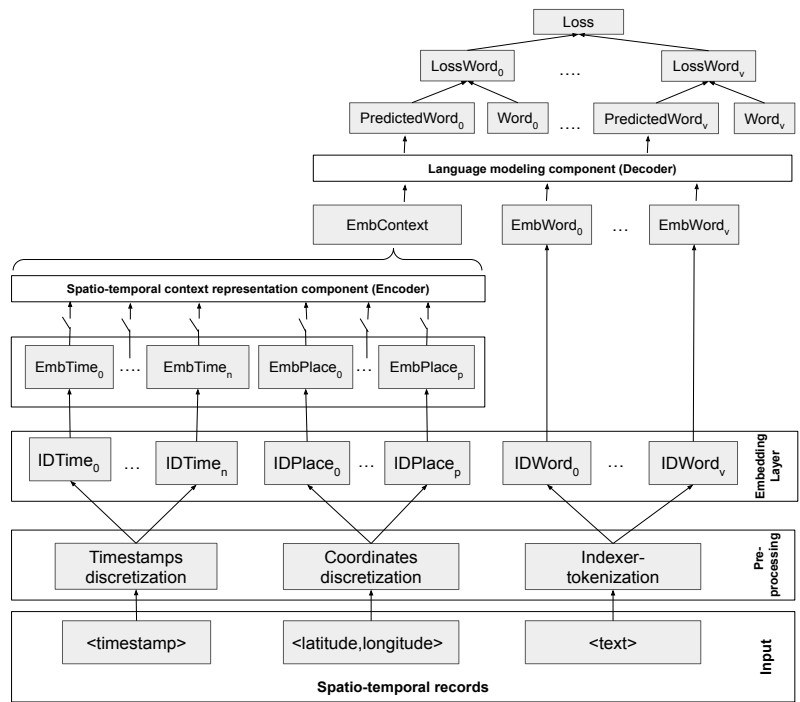

**Figure 1.** Model's architecture.

After the discretization step, the next step is building the spatio-temporal context (Equation (3)). Each timestamp can be discretized into $n$ temporal windows, and each coordinate can be discretized into $p$ spatial cells. The $n + p$ temporal windows and spatial cells represent the spatio-temporal context. Afterward, the context is passed through an Encoder layer that results in a context-representation tensor (EmbContext). This context-representation tensor is of invariant/fixed dimensions (<1,d> where d is the representation dimension) no matter how the context is selected. The EmbContext tensor is concatenated as the first element to the sequence of word embeddings (Equation (4)); this sequence [EmbContext, EmbWords] is passed through a Decoder that represents the language model. Finally, we compute the loss to minimize using as loss function the cross-entropy between the predicted sequence of words and the observed sequence of words in the training examples (Equation (5)). This is the general architecture that we propose. The main building blocks of our architecture (Encoder, Decoder) can be implemented using different approaches, such as recurrent neural networks or self-attention transformer blocks. We experiment with them in Section 4.

A salient property of our architecture is that it allows for representing time and space at different levels of granularities. This is achieved by modeling the spatio-temporal context as a sequence of discrete tokens that represent the particular semantics of each context type. For example, we could represent the temporal context by the hour of the day (0–23), day of the week (Sunday to Monday), week of the month, and month of the year (January to December) and the spatial context by block, neighborhood, district, etc.

$$IDTime_1, \ldots, IDTime_n = DiscTime(\langle timestamp \rangle)$$
$$IDPlace_1, \ldots, IDPlace_p = DiscCoordinates(\langle latitude, longitude \rangle) \quad (1)$$
$$IDWord_1, \ldots, IDWord_s = TextIndexer(\langle text \rangle)$$

$$EmbTime_1^{1,d}, \ldots, EmbTime_n^{1,d} = IDTime_1, \ldots, IDTime_n$$
$$EmbPlace_1^{1,d}, \ldots, EmbPlace_p^{1,d} = IDPlace_1, \ldots, IDPlace_p \quad (2)$$
$$EmbWord_1^{1,d}, \ldots, EmbWord_p^{1,d} = IDWord_1, \ldots, IDWord_s$$

$$SeqContext^{n+p,d} = [EmbTime_1^{1,d}, \ldots, EmbTime_n^{1,d}, EmbPlace_1^{1,d}, \ldots, EmbPlace_p^{1,d}]$$
$$EmbContext^{1,d} = Encoder(SeqContext^{n+p,d}) \tag{3}$$

$$SeqPred^{n+p,d} = [EmbContext^{1,d}, EmbWord_1^{1,d}, \ldots, EmbWord_p^{1,d}]$$
$$PredictedWord^{seqlen,vocabsize} = Decoder(SeqContext^{n+p,d}) \tag{4}$$

$$Loss = CrossEntropy(PredictedWord^{seqlen,vocabsize}, CorrectWord^{seqlen,vocabsize}) \tag{5}$$

### 3.5. Timestamps and Geo-Coordinates Discretization

To discretize geo-coordinates and timestamps, we use equal-size squared cells in the case of the geo-coordinates and hand-crafted temporal windows in the case of the timestamps. For timestamp discretizations, we use human semantic arrangements of time, in particular the hour of the day (0–23), day of the week (Sunday to Monday), week of the month (first week to the fifth week), and month of the year (January to December). Figure 2 shows a hierarchy describing these discretizations. For spatial discretization, we use equal-size spatial cells using the spatial coordinates as metric space. Figure 3 shows a hierarchy describing the squared-cell discretizations.

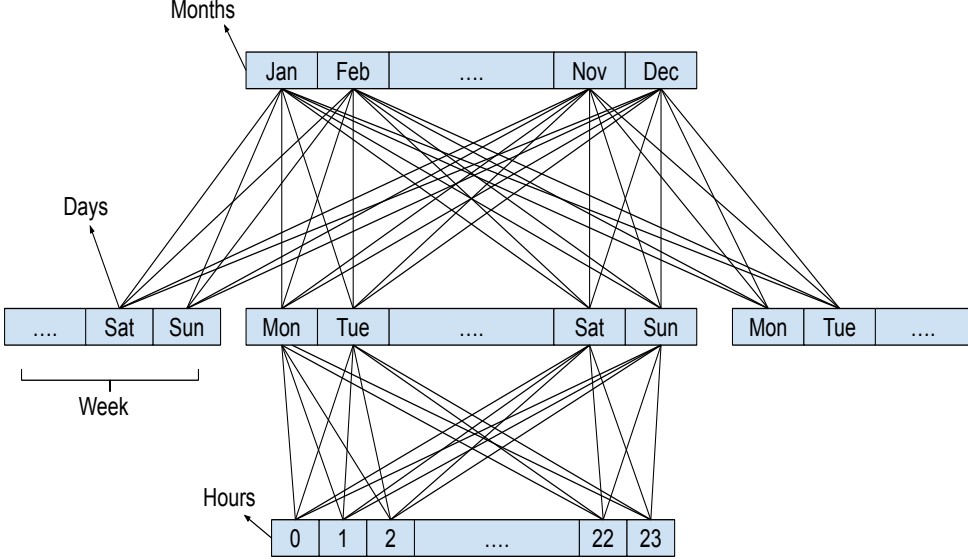

**Figure 2.** Hierarchy of timestamps discretization.

It is important to remark that our approach of representing contexts as discrete sequences allows for working at different levels of granularity. For example, a coarse representation could represent time by a single token corresponding to the month, where a more fine-grained approach could encode time as a sequence containing month, day, hour, etc. We argue that this is a core property of our architecture as it allows us to adapt the spatio-temporal context representation depending on the application. For example, for events related to daily activities (e.g., going to work, having lunch), granularities at the hour level should be more efficient. On the other hand, for events related to seasonal events (e.g., Christmas, holidays), month-level granularities should work better.

### 3.6. Parameters

In all our experiments, we use 128-dimensional embedding representation for *timestamp*, *location*, and *words*. The models are trained using mini-batch gradient descent with Adam optimizer [50]. We use 128 examples as batch size and early stopping on the validation dataset. We develop experiments with multi-layer GRU recurrent neural networks [45] and

Transformer-based neural networks for the Encoder/Decoder components of our proposed architecture. The GRU recurrent neural networks use a two-layer GRU with a hidden layer size of 128. Meanwhile, the Transformer-based neural networks are used in all cases also with 2 self-attention layers, 4 heads, and 128 vector size for queries, keys, and values (see [46] for additional details).

## 4. Experiments

In this section, we describe our experimental framework. The goal is to get a better understanding of the patterns that guide language generation in spatio-temporal contexts. In particular, looking at the data defined from tuples $\langle time, location, text \rangle$, the model will be evaluated in a traditional language modeling task (i.e., using the Perplexity metric). First, we describe the datasets. After that, we present the evaluation methodology. Then, we show the experimental results, and finally, we showcase studies of real-world applications of the studied models.

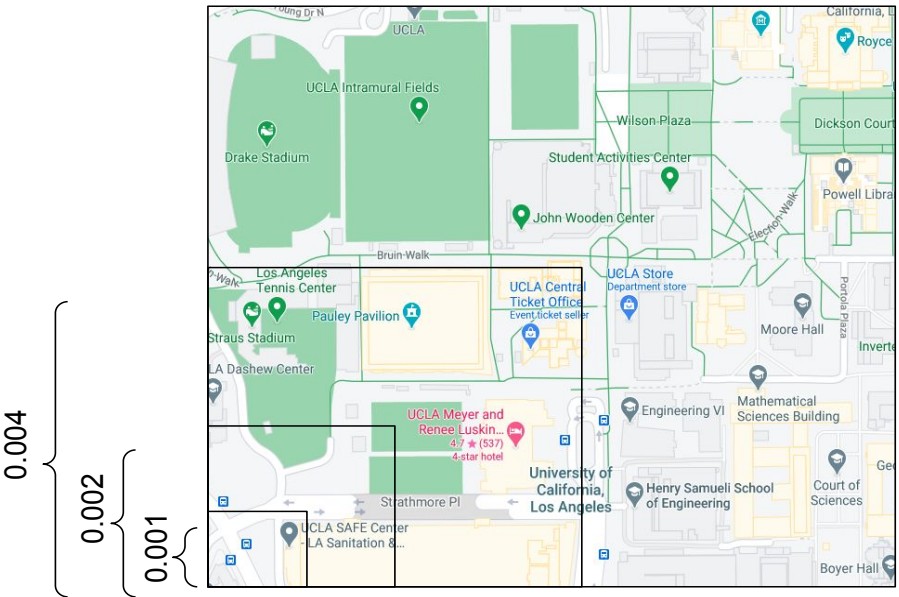

**Figure 3.** Hierarchy of coordinates discretization.

### 4.1. Datasets

We conduct experiments using two LBSN datasets: one from Twitter and other from Foursquare, each dataset is described next:

- Los Angeles ('LA-TW') : This dataset [10] is a set of geo-tagged tweets from Los Angeles, USA. It is 1,584,307 geo-tagged tweets from 2014.08.01 to 2014.11.30 (see Table 2).
- ('NY-FS'): This dataset was also first reported on [10]. It consists of Foursquare check-ins reported on Twitter by users in the city of New York, USA. The data contains 479,297 records check-ins from 2010.02.25 to 2012.08.16 (see Table 2).

**Table 2.** Datasets.

|  | LA-TW | NY-FS |
|---|---|---|
| Records | 1,188,405 | 479,297 |
| City | Los Angeles | New York |
| Start Date | 1 August 2014 | 25 February 2010 |
| End Date | 30 November 2014 | 16 August 2012 |

### 4.2. Evaluation Methodology

For each experiment, we split the dataset in training–validation–test, keeping 10% of each dataset as test, 10% for validation, and 80% for training. Given that the input to the models is a set of tuples in the form: $\langle timestamp, coordinates, text \rangle$, for each experiment, we set the vocabulary to the 12,288 most common words in the training set. The number of spatial cells and temporal windows is variable depending on the experiment. We filter out tuples where the number of words in the vocabulary is ten or less and reduce all URLs to the token "http".

Evaluation of language modeling is usually done using Perplexity [51]. Perplexity measures how well a language model predicts a test sample and captures how many bits are needed on average per word to represent the test sample. It is important to note that in Perplexity, the lower the score, the better the model. Perplexity, for a test set where all sentences are arranged one after other in a sequence of words $w_1, \ldots, w_T$ of length $T$, is defined as:

$$Perplexity = 2^{-\frac{1}{T} \log_2 p(w_1,\ldots,w_T)}. \tag{6}$$

### 4.3. Discretization Exploration

In order to better understand the spatio-temporal discretizations, in Figures 4 and 5, we show histograms of the timestamps and geo-coordinates discretizations for both datasets NY-FS and TW-LA. We show the 24 h of the day (0–23) and the discretization of geo-coordinates by (0.001 × 0.001) spatial cells.

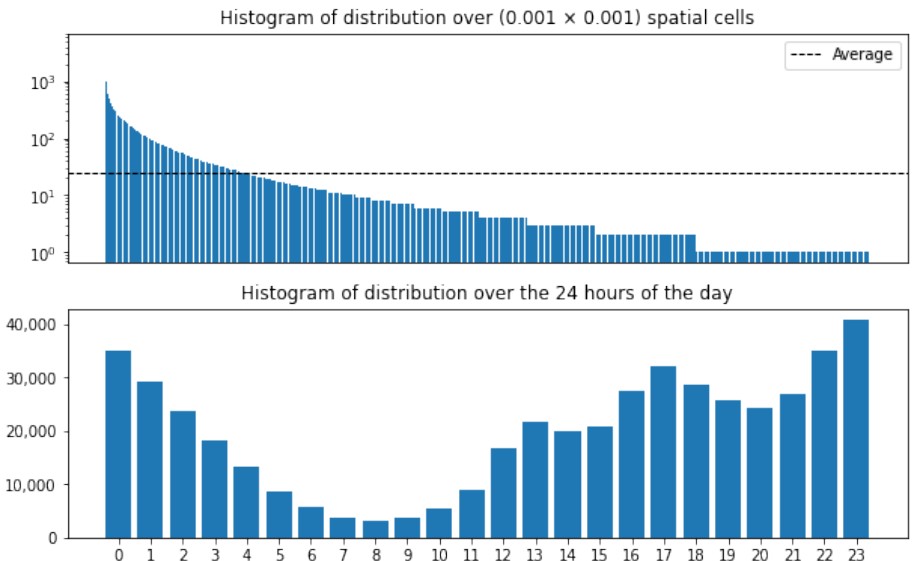

**Figure 4.** Histograms of distribution for the NY-FS dataset.

We can observe that for both datasets, early morning hours are the least frequent, starting to increase in the afternoon until the night hours. In total, there are 19,157 spatial cells for the NY-FS dataset and 84,693 for the LA-TW dataset. In the case of the NY-FS dataset, around 82% (15,796) of the cells have less than the average number of messages per cell (dotted line in Figure 4), while for the LA-TW, the distribution is similar: around 83% (70,529) of the cells have less than the average number of messages per cell (dotted line in Figure 5). These similarities in the patterns observed in the histograms indicate that even when these datasets were collected from different cities and in different time windows, there are patterns for text generation under spatio-temporal contexts that prevail independently of the place and time window in which the data were collected.

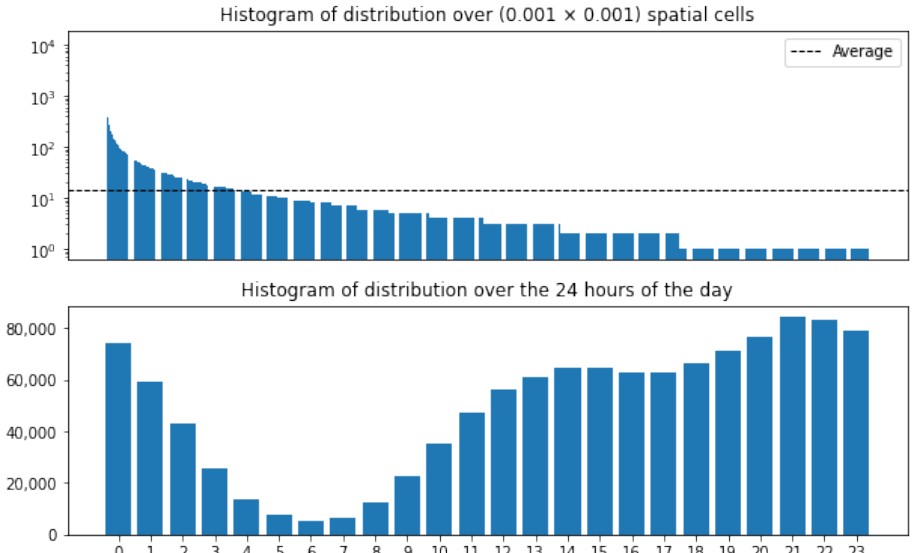

**Figure 5.** Histograms of distribution for the LA-TW dataset.

*4.4. Encoder–Decoder Analysis*

In our first set of experiments, we evaluate different options for the spatio-temporal context representation component (Encoder) and the language modeling component (Decoder) (see Section 3.4). In each case, we test two variants. For the Encoder, we test (1) projecting the embeddings output of the embedding layer with a fully-connected layer on top and (2) the Self-Attention Encoder representation proposed in [46] (without the positional encoding, since the order is irrelevant in the sequence of tokens representing the spatio-temporal context) also with a fully connected layer on top. For the Decoder, we test (1) a two-layer GRU recurrent neural network [45] and (2) a transformer-based two-layer Decoder representation proposed in [46].

In Table 3, we show the results for Foursquare and in Table 4, we show the results for Twitter. For both datasets, we test two different options for times and places in the Encoder: all times (alltimes), all places (allplaces), and all times–places (all). We can see that for both datasets and for each option of times and places, using only the embeddings in the Encoder performed better than using the Self-Attention component. While for the Decoder, the Self-Attention component performed equally better than the GRU in the same analysis. The combination Encoder(Embeddings)–Decoder(Self-Attention) achieved the best results in all cases. Our interpretation of these results is that the Self-Attention mechanism in the spatio-temporal context introduces noise between the units in the spatio-temporal context, while using only the Embeddings keeps the representations of the spatio-temporal units independent from each other. In the case of the Decoder, there is no such issue: what we are modeling is the sequential structure of the text that can be captured with the Self-Attention Decoder. In the next section, where we analyze different granularities for time and space, we use this setting of Encoder (Embeddings) and Decoder (Self-Attention) as the evaluation setting.

**Table 3.** Perplexity results for the Foursquare dataset from New York, testing only Embeddings and Self-Attention for the Encoder component and GRU-RNN or Self-Attention for the Decoder. In the "Context" column, h means hour, d means day of the week, w means week in the month, and m means month in the year. In addition, p1, p2, p4, and p8 mean squared cells of side 0.001, 0.002, 0.004, and 0.008.

| Context | Encoder | Decoder | Dataset | Perplexity |
|---|---|---|---|---|
| [] | - | GRU | NY-FS | 10.49 |
| [] | - | Self-Attn | NY-FS | 9.13 |
| [hdwm]-alltimes | Embeddings | GRU | NY-FS | 10.02 |
| [hdwm]-alltimes | Embeddings | Self-Attn | NY-FS | 9.00 |
| [hdwm]-alltimes | Self-Attn | GRU | NY-FS | 10.14 |
| [hdwm]-alltimes | Self-Attn | Self-Attn | NY-FS | 47.15 |
| [p1p2p4p8]-allplaces | Embeddings | GRU | NY-FS | 6.51 |
| [p1p2p4p8]-allplaces | Embeddings | Self-Attn | NY-FS | 5.45 |
| [p1p2p4p8]-allplaces | Self-Attn | GRU | NY-FS | 10.13 |
| [p1p2p4p8]-allplaces | Self-Attn | Self-Attn | NY-FS | 36.62 |
| [hdwm p1p2p4p8]-all | Embeddings | GRU | NY-FS | 6.38 |
| [hdwm p1p2p4p8]-all | Embeddings | Self-Attn | NY-FS | 5.34 |
| [hdwm p1p2p4p8]-all | Self-Attn | GRU | NY-FS | 10.14 |
| [hdwm p1p2p4p8]-all | Self-Attn | Self-Attn | NY-FS | 34.93 |

**Table 4.** Perplexity results for the Twitter dataset from Los Angeles. Testing only Embeddings and Self-Attention for the Encoder component and GRU-RNN or Self-Attention for the Decoder. In the "Context" column, h means hour, d means day of the week, w means week in the month, and m means month in the year. In addition, p1, p2, p4, and p8 mean squared cells of side: 0.001, 0.002, 0.004, and 0.008.

| Context | Encoder | Decoder | Dataset | Perplexity |
|---|---|---|---|---|
| [] | - | GRU | LA-TW | 63.03 |
| [] | - | Self-Attn | LA-TW | 57.35 |
| [hdwm]-alltimes | Embeddings | GRU | LA-TW | 61.90 |
| [hdwm]-alltimes | Embeddings | Self-Attn | LA-TW | 56.67 |
| [hdwm]-alltimes | Self-Attn | GRU | LA-TW | 63.02 |
| [hdwm]-alltimes | Self-Attn | Self-Attn | LA-TW | 193.77 |
| [p1p2p4p8]-allplaces | Embeddings | GRU | LA-TW | 61.13 |
| [p1p2p4p8]-allplaces | Embeddings | Self-Attn | LA-TW | 54.30 |
| [p1p2p4p8]-allplaces | Self-Attn | GRU | LA-TW | 62.42 |
| [p1p2p4p8]-allplaces | Self-Attn | Self-Attn | LA-TW | 161.14 |
| [hdwm p1p2p4p8]-all | Embeddings | GRU | LA-TW | 58.88 |
| [hdwm p1p2p4p8]-all | Embeddings | Self-Attn | LA-TW | 53.85 |
| [hdwm p1p2p4p8]-all | Self-Attn | GRU | LA-TW | 63.06 |
| [hdwm p1p2p4p8]-all | Self-Attn | Self-Attn | LA-TW | 72.80 |

*4.5. Spatio-Temporal Granularities Analysis*

In this section, we study how modeling time and space at different granularities influences the spatio-temporal conditioned language models. In Table 5, we show the results for the Twitter dataset from Los Angeles. We can see that in every case, including a spatial context or a temporal context improved the Perplexity results. In addition, the improvements for temporal contexts were marginal when compared to a language model that ignores the spatio-temporal context (first row in the table). The spatial contexts show notable improvements in all cases more than the temporal contexts; the larger the spatial cell, the better the results.

**Table 5.** Perplexity results for the Twitter dataset from Los Angeles using the combination Encoder(Embeddings)–Decoder(Self-Attention). In the "Context" column, h means hour, d means day of the week, w means week in the month, and m means month in the year. In addition, p1, p2, p4, and p8 mean squared cells of side: 0.001, 0.002, 0.004, and 0.008.

| Context | Cells | Dataset | Perplexity |
|---|---|---|---|
| [] | - | LA-TW | 57.35 |
| [h]—hour | 24 | LA-TW | 57.07 |
| [d]—day | 7 | LA-TW | 57.17 |
| [w]—week | 5 | LA-TW | 57.13 |
| [m]—month | 12 | LA-TW | 56.95 |
| [hdwm]—all times | 48 | LA-TW | 56.67 |
| [p1]—0.001 | 77,065 | LA-TW | 54.65 |
| [p2]—0.002 | 34,284 | LA-TW | 52.91 |
| [p4]—0.004 | 11,359 | LA-TW | 51.45 |
| [p8]—0.008 | 3283 | LA-TW | 51.30 |
| [p1p2p4p8]—allplaces | 125,992 | LA-TW | 54.30 |
| [hdwm p1p2p4p8]—all | 126,036 | LA-TW | 53.85 |

As a complement to the results in Table 5, in Table 6, we show the results with bigger spatial cells. We can see that instead of getting better results, Perplexity gets worst, which indicates that the sweet point to get the best results is with spatial cells between 0.008 and 0.016.

**Table 6.** Perplexity results for the Twitter dataset from Los Angeles using the combination Encoder(Embeddings)–Decoder(Self-Attention). In this case with squared cells of side: 0.016, 0.024, and 0.032.

| Context | Cells | Dataset | Perplexity |
|---|---|---|---|
| [] | - | LA-TW | 57.35 |
| [p]-0.016 | 1253 | LA-TW | 52.39 |
| [p]-0.024 | 460 | LA-TW | 52.81 |
| [p]-0.032 | 197 | LA-TW | 53.32 |

In Table 7, we show the results for the Foursquare dataset from New York. The Perplexities for this dataset are lower than the Perplexities for the Twitter dataset from Los Angeles. This is due to the fact that most of the Foursquare reports are generic texts generation suggested by the application. These texts only differ in most of the cases on the place that is checked in, while the Twitter dataset is mostly free texts. As for the spatio-temporal modeling, we observe similar results to the Twitter dataset; in all cases, including the spatio-temporal context improves the Perplexity. With the temporal contexts, producing marginal improvements, while the spatial contexts show the biggest margin in improvements. Contrary to the results over the Twitter dataset; with this dataset, a smaller cell size produced better results than the wider ones. We consider that this is due to texts being correlated to places of interest where people report activities in Foursquare (restaurants and small businesses) with a fine granularity.

**Table 7.** Perplexity results for the Foursquare dataset from New York using the combination Encoder(Embeddings)–Decoder(Self-Attention). In the "Context" column, h means hour, d means day of the week, w means week in the month, and m means month in the year. In addition, p1, p2, p4, and p8 mean squared cells of side: 0.001, 0.002, 0.004, and 0.008.

| Context | Cells | Dataset | Perplexity |
|---|---|---|---|
| [] | - | NY-FS | 9.13 |
| [h]—hour | 24 | NY-FS | 8.97 |
| [d]—day | 7 | NY-FS | 9.10 |
| [w]—week | 5 | NY-FS | 9.21 |
| [m]—month | 12 | NY-FS | 9.09 |
| [hdwm]—alltimes | 48 | NY-FS | 9.00 |
| [p1]—0.001 | 17,929 | NY-FS | 5.40 |
| [p2]—0.002 | 11,260 | NY-FS | 5.74 |
| [p4]—0.004 | 6060 | NY-FS | 6.10 |
| [p8]—0.008 | 3283 | NY-FS | 6.63 |
| [p1p2p4p8]—allplaces | 38,532 | NY-FS | 5.45 |
| [hdwm p1p2p4p8]—all | 38,580 | NY-FS | 5.34 |

As a complement to the results in Table 7, in Table 8, we show the results with smaller spatial cells. We can see that the results improve, and Perplexity gets lower. We could not continue the decrease in the spatial cell size because of resources restriction. In addition, in order to find a point where the Perplexity begins to deteriorate, we need to test spatial cells smaller than the regular size of popular places where activities are reported on Foursquare.

**Table 8.** Perplexity results for the Foursquare dataset from New York using the combination Encoder(Embeddings)–Decoder(Self-Attention). In this case with squared cells of side: 0.00075, 0.00050, and 0.00025.

| Context | Cells | Dataset | Perplexity |
|---|---|---|---|
| [] | - | NY-FS | 8.31 |
| [p]—0.00075 | 21,250 | NY-FS | 5.33 |
| [p]—0.00050 | 26,431 | NY-FS | 5.22 |
| [p]—0.00025 | 35,091 | NY-FS | 5.07 |

*4.6. Qualitative Analysis*

In this section, we perform a qualitative analysis of language generation for the studied models. First, we show examples of texts generated after training a spatio-temporal conditioned language model given a spatio-temporal context. Finally, we show Figures 6–8, where we can see attention weights that the text generation component gives to the elements in the spatio-temporal context. Attention weights can be particularly useful for the GIS community in our model, since they relate words to spatial and temporal contexts and offer interpretability. We can see the direct relationship between individual words and different granularities of representation.

In Table 9, we show examples of a language model trained with the Twitter dataset from Los Angeles with all granularities of time and space discretization (last row in Table 5). We selected two hubs for urban activities in Los Angeles: the Staples Center and Venice Beach. For the Staples Center, we selected a concert date of the British band the Arctic Monkeys and a date of a basketball game between the Los Angeles Lakers and the Los Angeles Clippers. We can observe that even for the same location, the texts generated can be associated with different events. For the examples using Venice Beach as the context, we can see that the generated texts are associated with beach activities.

This type of analysis shows the utility of the spatio-temporal conditioned language models trained over LBSN datasets to characterize human activities in urban areas. Figures 6–8 show examples given the Staples Center as the context. In Figure 6, we show a date from a Los Angeles Lakers game. We can see that the word "staples" is associated

with the finer granularity of geo-coordinates discretization, while the word "night" plays attention to the timestamp discretization as the hour of the day. In Figure 6, we show a date from a Katy Perry concert. We can see how the words "katyperry" and "at the staples center" are associated with the finest granularities of geo-coordinates discretization; meanwhile, the word "tonight", a more general term, is associated with the coarsest granularity. In Figure 8, we show an example with the geo-coordinates of Venice Beach as the spatial context. We can observe how the word "venice" is associated with the finest level of spatial discretization; while the word "beach" is associated with the second finest granularity, "beach" is a more general term than "venice", but it also is only associated with coastal regions in a city.

**Table 9.** Examples of text generation after training a spatio-temporal conditioned language model with the dataset of Twitter from Los Angeles. This table shows results for two points of interest: the Staples Center and Venice Beach. For the Staples Center, we selected a date of a concert and a date of a basketball game.

| Context | Text Generated |
| --- | --- |
| (Staples Center) (34.043; −118.267) (Concert Date) '7 August 2014 22:00:00' | ['<START>', 'taking', 'a', 'break', 'from', 'the', 'arctic', 'monkeys', 'concert', 'and', 'i', 'love', 'the', 'place', 'if', 'you', 'are', 'here', '#staples', 'staplescenter', 'http', '<END>' ['<START>', 'during', 'the', 'night', '#arctic-monkeys', 'http', '<END>'] ['<START>', 'arctic', 'monkeys', 'anthem', 'with', 'my', 'mom', 'at', 'staples', 'center', 'http', '<END>'] |
| (Staples Center) (34.043; lon = −118.267) (Game Date) '31 October 2014 22:00:00' | ['<START>', 'just', 'posted', 'a', 'photo', '105', 'east', 'los', 'angeles', 'clippers', 'game', 'http', '<END>'] ['<START>', '#lakers', '#golakers', 'los', 'an-geles', 'lakers', 'surprise', 'summer', '-', 'great', 'job', '-', 'lakers', 'nation', 'http', '#sportsroadhouse', '<END>'] ['<START>', 'who', 'wants', 'to', 'go', 'to', 'the', 'lakings', 'game', 'lmao', '<END>'] |
| (Venice Beach) (33.985; −118.472) (Date) '24 August 2014 13:50:00' | ['<START>', 'touched', 'down', 'venice', 'beach', '#venice', '#venicebeach', 'http', '<END>'] ['<START>', 'venice', 'beach', 'cali', '#nofil-ter', '#venice', '#venicebeach', 'is', 'rolling', 'great', '<END>'] ['<START>', 'who', 'wants', 'to', 'go', 'to', 'venice', 'beach', 'shot', 'on', 'the', 'beach', '<END>'] ['<START>', 'venice', 'beach', '#venice-beach', '#california', '#travel', 'venice', 'beach', 'ca', 'http', '<END>'] ['<START>', '#longbeach', '#venicebeach', '#venice', '#beach', '#sunset', '#venice', '#venicebeach', '#losangeles', '#california', 'http', '<END>'] |

The above examples illustrate the potential of our model for spatio-temporal analyses. On the one hand, we demonstrate that our language models are able to generate sentences that efficiently and coherently describe a spatio-temporal context. This can be especially useful for researchers trying to describe or summarize an event using natural language from spatio-temporal contexts. Moreover, our attention weights provide an interpretable relationship between text, space, and time. To the best of our knowledge, this is the first

work to use an attention mechanism for this purpose. These interpretations are valuable, as they provide insights into how space and time influence what people say (whether on social networks or any other data source of this nature). Although neural networks are known to be difficult to interpret, attention weights are a well-known example of an interpretable component that has been widely used in machine translation and video captioning, among others. We hope that the results presented here will increase interest in the use of this mechanism in spatio-temporal domains.

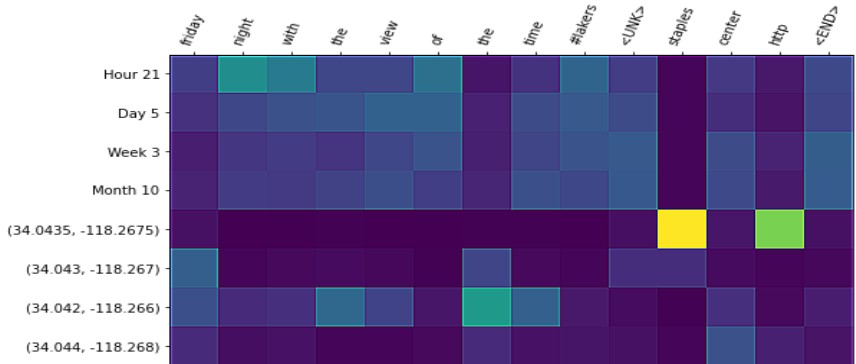

**Figure 6.** Example sentence attention to the spatio-temporal context. Yellow means more attention, while blue means less attention. In this case, with the STAPLES Center as context in a Los Angeles Lakers game day.

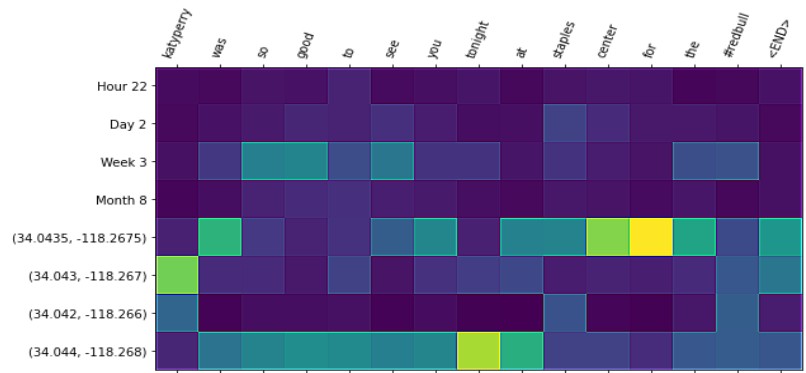

**Figure 7.** Example sentence attention to the spatio-temporal context. Yellow means more attention, while blue means less attention. In this case, with the STAPLES Center as context in a Katy Perry concert day.

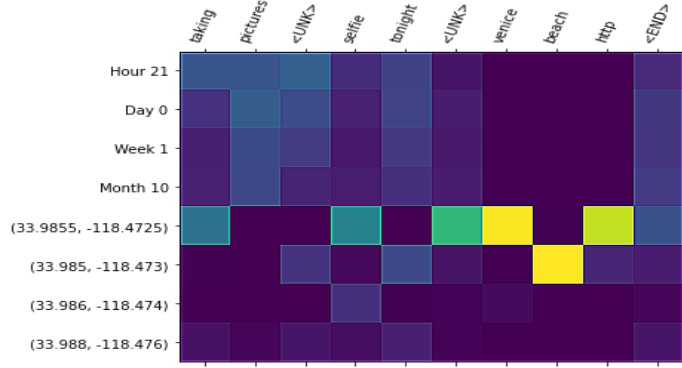

**Figure 8.** Example sentence attention to the spatio-temporal context. Yellow means more attention, while blue means less attention. In this case, with Venice Beach at night as context.

## 5. Conclusions

In this work, we studied the problem of modeling spatio-temporal annotated textual data. We studied how different granularities of time and space influence spatio-temporal conditioned language generation on location-based social networks. We proposed a neural language model architecture adaptable to different granularities of time and space. A remarkable result of our experiments over two datasets from social networks Twitter (Los Angeles) and Foursquare (New York) is that each dataset has its own optimal granularity setting for spatio-temporal language generation. Since our proposed architecture is adaptable to modeling time and space at different granularities, it is capable of capturing patterns according to each dataset. These results directly answer our research question by empirically demonstrating that an appropriate adjustment of temporal and spatial granularities can benefit spatio-temporal language modeling/generation. On our qualitative evaluations, first, we show how the proposed model can be used to summarize activities in urban environments with natural language generation. This application highlights the importance of modeling the sequential structure of texts in order to generate coherent descriptions for spatio-temporal contexts. Secondly, we show how words with distinct semantics are linked to spatial cells and temporal windows related to their semantics.

We foresee valuable future research opportunities by working with more recent datasets and with the use of handcrafted discretizations. We chose to conduct our experiments with these datasets in order to keep the evaluation process consistent with previous works. For the timestamp and geo-coordinates discretizations, we would like to avoid the use of hard delimitations between cells, as this can lead to times and places that may be close to each other being assigned to different cells.

**Author Contributions:** Conception and design of study: Juglar Diaz, Barbara Poblete and Felipe Bravo-Marquez; acquisition of data: Juglar Diaz; analysis and/or interpretation of data: Juglar Diaz, Barbara Poblete and Felipe Bravo-Marquez. Drafting the manuscript: Juglar Diaz; revising the manuscript critically for important intellectual content: Juglar Diaz, Barbara Poblete and Felipe Bravo-Marquez. Approval of the version of the manuscript to be published: Juglar Diaz, Barbara Poblete and Felipe Bravo-Marquez. All authors have read and agreed to the published version of the manuscript.

**Funding:** This work was supported by Millennium Institute for Foundational Research on Data (IMFD), FONDECYT Grant No. 1191604, FONDECYT Grant No. 11200290 and CONICYT-PCHA/Doctorado Nacional/2016-21160142.

**Institutional Review Board Statement:** Not applicable.

**Informed Consent Statement:** Not applicable.

**Data Availability Statement:** In this work, we use two datasets: A dataset of geo-tagged tweets from Los Angeles and a dataset of Foursquare check-ins from New York. Both datasets were first reported in [10]. We downloaded the datasets from the link provided by the authors in (https://drive.google.com/file/d/0Byrzhr4bOatCck5mdWJnb2xudVk/view?resourcekey=0-4JNsJjEN-EmRX9qO1dajkw, accessed on 30 October 2021). and created our pre-processed versions that can be found in (https://drive.google.com/drive/folders/19pN-U3jJZxWZgVYkGFyso2Bs9cLs-631?usp=sharing, accessed on 30 October 2021).

**Conflicts of Interest:** The authors declare no conflict of interest.

## Abbreviation

The following abbreviations are used in this manuscript:

LBSN    Location-based social networks

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
