# Peer review of "Language Modeling on Location-Based Social Networks"

_ijgi, doi:10.3390/ijgi11020147_

Round 1

Reviewer 1 Report

In this work, the authors propose to combine spatial and temporal information to predict words. The general framework is understandable, but there are some major issues:
1. As there are a number of existing methods in Table 1, can the authors compare with them in terms of perplexity in the experiments? 
2. Why the major contribution contains “an evaluation framework”? Isn’t it evaluated by Perplexity? And what is w in Eq. 1?
3. The abstract may highlight the major contribution of this work, comparing it with existing spatio-temporal text prediction methods. 
4. In Fig 1, how EmdTime and EmbPlace are fed into Encoder, by a fully connected layer? The model should be more accurately described by equations. 
5. In Fig 3, 0.001 is in kilometer? If so, a 10*10km area contains 10^4*10^4 blocks on average, much more than the data items (10^6). If so, the data size may not be enough, or are the data items concentrated on some blocks? The authors may provide a histogram to illustrate the spatial-temporal distribution of the data items. 

Reviewer 2 Report

Content
----------
The goal of this paper is to propose an evaluation framework agnostic to how time and space are modeled in conjunction with text and a neural language model architecture adaptable to different granularities of time and space. The main contribution of this research is that define the model on spatio-temporal annotated textual data. The experimental results showed that how an attention based neural network can be used to obtain information about patterns that guide language generation in spatio-temporal contexts.

Major comments
--------------
1.  Abstract
The abstract is a little bit lengthy.

Evaluation
--------------

This is a well written article.
Given the above, I'm in a position to accept. 

Reviewer 3 Report

Dear Authors, Thank you very much for submitting your paper to ISPRS Int. J. Geo-Inf.  I read it with great interest. I must emphasize that your article has several advantages. These are certainly: the well-thought-out structure of the study, the clarity and transparency of the statements and the consistent narrative. Another plus is the literature review leading to the demonstration of contributions related to identifying academic gaps. Another advantage is the development of a model which has some potential for application in practice (additionally, it complements the literature on the subject). Unfortunately, however, I also observed some flaws which speak against the publication of your paper.  Among the minor ones, I could point out editing errors (double spaces and incorrect arrangement of tables/graphs, which disturbs the statement's logic).  I would also recommend more prominence for the identified academic gaps (which now include "contributions").  In section 3.3. you announce the formulation of the research question, but it does not appear (there is only a hypothesis).  It would also be good if, in section 3.4, you could explain why you refer to neutral networks (especially as your model is embedded in them).  In the "Conclusions" section, you do not point out the research limitations at all (and there are quite a few). Additionally, you do not address the research question and hypothesis. Instead, you repeat the content relating to contributions. Meanwhile, it would be worthwhile to significantly expand them, demonstrating how specifically your analyses would contribute to improving the practical solutions mentioned in lines 812-817. I do not find in your research references to, e.g. urban development, crime reports, etc. They appear at the article's beginning and end, but in the research itself, you do not prove how the results obtained could contribute to implementing the changes you mention.  In addition, I think that section 4.5 in its current form is redundant. It is too cursory, short, in-depth and random. I would suggest that this text focus solely on the quantitative analyses that demonstrate how the model works and move the qualitative studies to a separate text. However, I have most comments and questions about Section 4.  It would be helpful to justify why you chose only TW and FS and these two cities (LA and NY) for analysis. I would also like to know in what time frame did you conduct the study? My biggest concern is that the data used for the analysis is from 2010-2014 (table 2). I think they are outdated, considering the development of technology, social networks, and their functionalities, not to mention the pandemic's changes.  I am very sorry, but in my opinion, a paper focusing on new technologies and announcing theoretical and practical implications cannot be based on data from almost a decade ago. The solution would be to either replace the old data with new data or add new data to the old data and compare them (this would give us a fascinating comparative study to check your model's performance over the years). I hope you decide to make changes and update your text, the theoretical part of which is promising.  For the time being, however, the paper does not qualify for publication, although it is worth working on its shape to apply for publication in the future. 

Round 2

Reviewer 3 Report

Dear Authors, 

Thank you very much for sending the resubmitted version of your paper. As I stated previously, it already had some advantages. However, the long list of flaws made the article necessary to revise. 

I have read the new version of the text, and I also got familiar with the cover letter. I am pleased that you have decided to improve your work quality and refer to each of my doubts. Your answers are well-thought, reliable, balanced and very substantive. You explain your decision by giving solid and convincing arguments (even regarding the data set).

You have also changed the paper significantly in every aspect I have referred to. Now I can confirm that is it almost ready for publishing. I would like to ask you to check the language once more - there are some editing errors left (i.e. double spaces, numerations issues).

Congratulations.